A worldwide comparison of long-distance running training in 2019 and 2020: associated effects of the COVID-19 pandemic

Afonseca Leonardo A.
Watanabe Renato N.
Duarte Marcos duartexyz@gmail.com
Biomedical Engineering, Universidade Federal do ABC , Sao Bernardo do Campo , SP , Brazil
Clemente Filipe
Electronic publication date: 2022 Mar 25
Publication date: 2022
Volume: 10
Electronic Location ID: e13192
Received 2021 Oct 14; Accepted 2022 Mar 8
Copyright: ©2022 Afonseca et al.
Copyright year: 2022
Copyright holder: Afonseca et al.
License: This is an open access article distributed under the terms of the Creative Commons Attribution License, which permits unrestricted use, distribution, reproduction and adaptation in any medium and for any purpose provided that it is properly attributed. For attribution, the original author(s), title, publication source (PeerJ) and either DOI or URL of the article must be cited.
License URL: https://creativecommons.org/licenses/by/4.0/

Keywords: Physical activity, Sports, Public health, Data science, Running

Funding: Fundação de Amparo à Pesquisa do Estado de São Paulo from Brazil 2015/14810-0 This work was supported by Fundação de Amparo à Pesquisa do Estado de São Paulo from Brazil (2015/14810-0). The funders had no role in study design, data collection and analysis, decision to publish, or preparation of the manuscript.

==============================
Objective

The goal of the present study was to investigate possible effects of the COVID-19 pandemic on long-distance running training.

Methods

This is a retrospective study with a within-subject design. We analyzed 10,703,690 records of running training during 2019 and 2020, from 36,412 athletes from around the world. The records were obtained through web scraping of a large social network for athletes on the internet. A potential long-distance runner was defined as a user of the social network who had a record of running at least one of the six World Marathon Majors by 2019.

Results

In 2020, compared with 2019, in total there was a 3.6% decrease in the number of athletes running, a 7.5% decrease in the distance and 6.7% in the duration of running training. There were large variations in these variables throughout 2020, reaching 16% fewer athletes running weekly and 35% lower running distance (Cohen’s d = 0.34, p < 0.001) and 33% lower running duration (Cohen’s d = 0.30, p < 0.001) in September 2020. The beginning of the decrease in running training in the first quarter of 2020 coincides with the beginning of the adoption of measures to restrict the COVID-19 pandemic; but as of the second quarter of 2020, running training appears to have undergone variations unrelated to the preventive measures. Among the ten most represented countries in the dataset, running training in Brazil appears to have been the most affected by the COVID-19 pandemic and restriction measures.

Conclusion

The wide variations in long-distance running training throughout 2020 are likely related to the COVID-19 pandemic. As for the total volume, the observed decreases of up to 7.5% in the outcome variables related to running training in 2020 could also be attributed to the COVID-19 pandemic, but other factors such as injury, illness or lack of interest, may also have contributed to these decreases.

Introduction

The Coronavirus Disease 2019 (COVID-19) had a profound impact on the lives of people worldwide in 2020, and continues to impact lives in 2021. As of December 2021, approximately 276 million cases have been confirmed and 5.4 million deaths have been attributed to COVID-19 (World Health Organization, 2021). Physical activity is essential to achieve good health and wellbeing (White et al., 2019; Cunningham et al., 2020), particularly when meeting physical activity guidelines associated with a reduced risk for severe COVID-19 outcomes (Sallis et al., 2021).

There is a growing body of evidence suggesting that overall physical activity decreased after March 2020, when preventive measures were adopted against COVID-19, such as social distancing and isolation. A systematic review of 66 scientific articles on this subject (total number of subjects, N = 86,981) reported a decrease in the practice of physical activity during the pandemic (Stockwell et al., 2021). Of these 66 studies, only four (including a total of 468 subjects) used device-based measures (using a device with an accelerometer or GPS); the other 62 studies relied on questionnaires. Another study, not included in the aforementioned review, quantified physical activity in the first half of 2020 using data from 455,404 users of the Argus app (from the company Azumio) for smartphones with accelerometers and also reported a decrease in the practice of physical activity (Tison et al., 2020). However, two recent studies, which specifically investigated running, reported increased running volume during the pandemic (DeJong, Fish & Hertel, 2021; Chan et al., 2022). Based on data from tens of millions of users, the companies Fitbit and Strava, which monitor user’s physical activity using their own trackers, smartphones, or smartwatches with accelerometers or GPS, reported on their blogs that they observed a decline in physical activity in the first half of 2020 compared with 2019 (Fitbit, 2020; Strava, Inc 2020). Strava did see a general increase in physical activity when considering the entire period analyzed in 2020 (until the month of October), but they also included new users in 2020 in their comparison (Strava, 2020). That is, this increase in physical activity could simply be because more people have joined the Strava app.

As the impact of the COVID-19 pandemic on the practice of physical activity is still a very recent topic, most scientific articles cited report data for an incomplete time window for the year 2020, and the only study with direct measures of physical activity by a large sample of subjects reports results for only the first half of 2020 (Tison et al., 2020). In the present study, our objective was to investigate how long-distance running training throughout the world and throughout 2020, quantified as a time series at a weekly frequency, might have been affected by the COVID-19 pandemic, compared with training in 2019. Comparing the total long-distance running training during all periods of 2019 and 2020 allowed us to better control for possible effects of seasonality. We decided to focus on long-distance running as an indicator of physical activity, as it is one of the most popular and accessible modalities for people seeking a healthy lifestyle (Stamatakis & Chaudhury, 2008; Hespanhol Junior et al., 2015). Our hypothesis was that compared to 2019, the training volume for long-distance running in 2020 would be reduced. A secondary objective of this study was to make the dataset with the running training records publicly available and, in this article, we describe how this data is organized and how it can be accessed.

Methods

This is a retrospective study with a within-subject design. This study used only data related to the practice of physical activity by users of the Strava app and who chose to publicly share their data on the Strava website (Strava Inc, 2020). Data extraction was performed using web scraping, a technique in which software is written to implement a web robot to automate the process of accessing and extracting specific data from a website and saving them for later analysis (Baskaran & Ramanujam, 2018; Zheng, Zhu & Lyu, 2018).

Ethics declarations

There was no direct involvement of humans as participants in this research. The data used in this study were extracted from public pages in the internet without contact with any of the Strava users; the data of interest that we gathered were anonymized into a final dataset and individual Strava users cannot be identified. Regarding the use of web scraping, it was necessary because of the large number of pages retrieved for this scientific research, which has no association with commercial interests. We scraped data from less than 40,000 users for a few days, at a low rate so as not to overload the Strava website (the company Strava reports having 73 million users Strava, Inc 2020). All procedures performed in this study were in accordance with the local Ethics Research Committee standards and with the 1964 Declaration of Helsinki and its subsequent amendments. This study was exempted by the Ethics Research Committee of the Federal University of ABC, Brazil.

Data gathering

We first searched for Strava users who were potential long-distance runners and then extracted all user activities and related information from January 2019 to December 2020. A potential long-distance runner was defined as a user who had a record of running at least one of the six World Marathon Majors (Abbott World Marathon Majors , 2020), i.e., one of the New York, London, Boston, Tokyo, Chicago, or Berlin marathons, by 2019. This delimitation was necessary because the Strava application has potentially millions of users and it would be impracticable to search all associated web pages. We excluded users who did not share their activities publicly and who did not provide information on gender and age group. Using these criteria, we have successfully identified a cohort of distance runners, who henceforth are referred to as athletes. After extraction, all data generated were merged into a single text file for each year, anonymized, and transformed so that they could be analyzed in an organized and structured way, henceforth referred to as a dataset. All the analyses were performed using computer programs written in Python and executed in the form of Jupyter Notebooks, a powerful, versatile, and friendly tool for data science (Kluyver et al., 2016).

In the dataset, each athlete’s age group was extracted from the Strava website with the most recent marathon segment in which the athlete participated. The age group was in reference to the year in which the marathon took place. This needed to be updated for 2019. To do so, we estimated the athlete’s initial age as the average of the reported age group interval. Then, the updated age group for 2019 was computed, adding to this estimated age the difference between 2019 and the year of the most recent marathon in which the athlete participated. Finally, the updated age was transformed into one of the existing age groups. For the present study, we used only three age groups: 18–34 (young), 35–54 (middle age), and 55 or more (older) years old. To determine the athlete’s country, we searched for up to 10 activities per athlete that had information about the GPS location of the activity. Then, we searched for the country corresponding to the most frequent location found, and defined this as the country of origin of the athlete.

Dataset description

The dataset we created and analyzed in this study includes records of 10,703,690 long-distance running activities by 36,412 athletes from 130 countries in 2019 and 2020. The data with the athletes’ activities are contained in dataframe objects (tabular data) and saved in the Parquet file format using the Pandas library, part of the Python ecosystem for data science. Each Pandas dataframe contains the following information, as different columns, for each athlete, as different rows (the first word identifies the name of the column in the dataframe):

datetime: date of the running activity (string);

athlete: a computer-generated ID for the athlete (integer);

distance: distance of running (floating-point number, in kilometers);

duration: duration of running (floating-point number, in minutes);

gender: gender (string ‘M’ of ‘F’);

age_group: age interval (one of the strings ‘18–34’, ‘35–54’, or ‘55+’);

country: country of origin of the athlete (string);

major: list of marathons the athlete ran (string).

For convenience, we created files with the athletes’ activities data sampled at different frequencies: day ‘d’, week ‘w’, and month ‘m’ (i.e., there are files with the distance and duration of running accumulated at each day, week, and month) for each year, 2019 and 2020. Accordingly, the files are named ’run_ww_<yyyy>_<f>.parquet’, where <yyyy>is ’2019’ or ‘2020’ and <f>is ‘d’, ‘w’, or ‘m’ (without quotes). In addition, the dataset also contains data with different government’s 2020 stringency indexes for the COVID-19 pandemic; these data are saved as text files in the dataset. The Jupyter notebooks made available as a Github repository exemplify how to handle these data.

Data analysis

The original dataset analyzed in this study included 37,595 athletes and 14,644,391 separate activity bouts performed in 2019 and 2020. To compare each athlete’s long-distance running training in 2019 with that of 2020, only athletes who ran at least once in 2019 and only running activities were selected from the dataset. We removed any new athletes that started on the Strava app in 2020, as we were interested in performing a within-subject analysis. These restrictions reduced the dataset to 10,703,690 running activities by 36,412 athletes. To quantify long-distance running training, we reported the following outcome variables in this study: the number of athletes who ran per week, the number of runs per week, the weekly running distance and duration (these two last variables are measures of running volume), and the weekly average pace (computed as the ratio between weekly running duration and distance). We defined the weekly running volume as the total distance covered in kilometers (or the total duration in minutes) per week, calculated by simply summing all distances (or durations) covered in the week. The total number of runners per week was calculated as the number of runners who each had a weekly running volume greater than zero. The number of runs per week for each athlete was calculated as the sum of days with a running volume greater than zero in the week. Because we organized the data based on whole weeks only, to use all available running registry data, we needed to adjust for extra days in each year that did not fit into a 7-day week. Specifically, the last week of each year was made up of 8 days for 2019 and 9 days for 2020. Then, we adjusted the data for these final weeks by multiplying the values of the outcome variables by 7/8 and 7/9, respectively. After this processing, the number of running athletes per week had 52 values per year and the variables for weekly running volume (distance and duration) had 52 values per year for each athlete. Finally, we calculated the relative difference (in %) between years for the outcome variables reported in this study: for the running volume variables, the difference between the running volume in 2020 and 2019 for each athlete at each week divided by the overall mean weekly running volume in 2019 and then multiplied by 100. In a similar way, the total number of athletes running per week was described by a relative difference (in %): the difference between the number in 2020 and in 2019 for each week divided by the mean number of running athletes per week in 2019. These measures of relative differences for the outcome variables were calculated considering the entire sample of all athletes and also by subgrouping the data by gender (female, male) and age range (18–34, 35–54, 55+).

In order to investigate the possible causes of variations in the weekly running volume and in the number of athletes running in 2020 versus 2019, we qualitatively compared these data with the relevant government’s stringency index (Ritchie et al., 2020). This 2020 stringency index was calculated as an average score (from 0 to 100) for each day and country for the following nine COVID-19 response metrics: school closings; closings of workplaces; cancellation of public events; restrictions on public meetings; public transport closures; requirements for staying at home; public information campaigns; restrictions on internal movements; and international travel controls. An average 2020 stringency index across countries was calculated, with the score for each country weighted by the number of athletes in that country in the dataset.

Statistical analysis

The number of athletes is only a single value per week (52 values per year) and no statistical tests are applied, but we also calculated the average number of athletes per week across the 52 weeks and statistical tests are calculated for this variable. The weekly running volume (distance and duration) differences were not normally distributed (revealed by visual inspection of the histogram, skewness and kurtosis values and the Jarque–Bera test); although the distribution is symmetric, there are excessive number of zeros and long tails in the distribution. The large number of zeros is due to the fact that a zero value was assigned to a given week whenever the athlete did not run that week. The long tails in the distribution are not due to outliers; they reflect the fact that an athlete ran a much longer distance (duration) than the average in a given week in one year and not the same week in the other year.

The weekly running volume (distance and duration) differences are summarized by mean, standard deviation, and corresponding 95% confidence interval for the mean (CI95%). Since the data distribution is not normal, but it is symmetric, the CI95% ranges were calculated by the non-parametric percentile bootstrap with 100,000 random samples with replacement of the same size as the original dataset (Efron & Tibshirani, 1993). Cohen’s d effect size for within-subject design was calculated as the mean difference divided by standard deviation for the data and the following conventional rule was adopted: d(0.01) = very small, d(0.2) = small, d(0.5) = medium, d(0.8) = large, d(1.2) = very large, and d(2.0) = huge. To test whether an effect was associated with the COVID-19 pandemic, we tested whether the means of the weekly running volume differences were different than zero (the null hypothesis is the mean of the population from which the data were sampled is equal to zero). To this end, we conducted non-parametric two-sided bootstrap hypothesis testing, in which the achieved significance level (the p-value) is the fraction of bootstrap samples at least as extreme as that actually observed under the null hypothesis (Efron & Tibshirani, 1993).

The initial significance level was set at 0.05. To adjust for the multiple comparisons performed (52 tests, one for each week), we applied the Šidák correction and the resulting significance level was set at 0.001. The outcome variables were plotted against week of the year. The corresponding mean values of weekly running volume and number of athletes running per year and the calculated average 2020 stringency index across countries are also shown in the graphs. The Jupyter notebooks we created and made available as Github repository contain the source code written to perform these analyses.

Results

Long-distance running training in 2019 and 2020

Of all 36,412 athletes in 2019 in the dataset, 35,083 ran at least once in 2020 (3.6% reduction). Descriptive and inferential statistics for the outcome variables number of athletes and of running days, running distance, duration, average pace per week across athletes are shown in Table 1. See graphs of the distributions of the weekly number of runs and running distance by quarter are shown in Figs. 1 and 2, respectively. Regarding demographic characteristics of the athletes, 76% were male; 34% were between 18 and 34 years old, 59% were between 35 and 54 years old, and the remaining 7% were 55 years old or older. The most common regions of origin of the athletes were: 43% from North America (38% from the United States), 44% from Europe (21% from the United Kingdom), 8% from Asia, and 3% from South America. Regarding the marathons that athletes participated in, 98% of them ran up to two Majors and 98% of the athletes ran a Major between the years 2014 and 2019. Only 67 athletes identified themselves as professionals on their web pages.

Table 1 Outcome variable statistics.

Mean and 95% confidence interval of the weekly outcome variables for the 36,412 athletes’ running data during 2019 and 2020. Cohens d (d) and p-value (p) of the null hypothesis significance testing for the relative difference are also shown.

Values per week	2019	2020	Difference (%)	Statistics: d, p	
Number of athletes	27404 (27103, 27683)	25612 (25222, 25970)	−6.5 (−7.9, −5.2)	1.2, <0.001	
Number of runs	2.46 (2.45, 2.48)	2.41 (2.39, 2.43)	−2.2 (−2.6, −1.7)	0.05, <0.001	
Distance (km)	29.2 (29.0, 29.5)	27.0 (26.8, 27.3)	−7.5 (−8.0, −7.0)	0.16, <0.001	
Duration (min)	160.5 (159.3, 161.8)	149.7 (148.4, 151.1)	−6.7 (−7.2, −6.2)	0.14, <0.001	
Pace (min/km)	5.85 (5.79, 5.95)	5.73 (5.65, 5.87)	−2.0 (−4.4, 0.7)	0.01, 0.116	

Figure 1 Histogram of the number of runs.

Distribution of the number of runs per quarter (Q1–Q4) in 2019 and 2020 for all athletes who ran at least once a year. The inset plot details the distribution in the tail (large number of runs). The width of the bin is four runs.

Figure 2 Histogram of the average weekly running distance.

Distribution of the the average weekly running distance per quarter (Q1–Q4) in 2019 and 2020 for all athletes who ran at least once a year. The inset plot details the distribution in the tail (long running distances) with a logarithmic scale in the vertical axis. The width of the bin is 10 km.

The annual time series for the number of athletes, running distance, and the differences for these variables between 2019 and 2020, normalized by the average values for 2019 for all athletes in the dataset are shown in Fig. 3. The average value of the government stringency index (Ritchie et al., 2020) throughout 2020 is also shown in Fig. 3. One can see that the relative differences in weekly number of athletes and running distance started to change in March, when a large decrease in running distance was observed in 2020 (a decrease of up to 24% in distance (d = 0.22, p < 0.001) in the third week of March), which coincided with the beginning of the preventive measures, as indicated by the 2020 stringency index. However, in the second quarter of 2020, the relative difference was in the opposite direction. The running distance increased in 2020 in this period, by up to 18% (d = 0.16, p < 0.001) in the third week of April. In addition, the running distance no longer displayed an association with the 2020 stringency index, which remained at a high level of restrictions until the end of the year. In the third quarter of 2020, the running distance fell again, this time more severely and for a longer time, reaching a minimum of 35% lower running distance (d = 0.34, p < 0.001) and 16% fewer athletes running weekly in the last week of September. As of the second half of the last quarter of 2020, the running volume was once again similar to that of 2019. The results for the running duration variable are similar and are not presented here, see the Jupyter Notebooks in the GitHub repository. The relative difference between the number of athletes running per week in 2019 and 2020 was qualitatively similar to the pattern of running volume, but in quantitative terms, the difference between the number of athletes running in 2019 vs. 2020 was half the difference between the running volume in 2019 vs. 2020.

Figure 3 Comparison between the weekly running training of 36,412 athletes worldwide over the years 2019 and 2020.

(A) Number of athletes running per week by year. (B) Mean and 95% confidence interval (95% CI) of weekly running distance by year. (C) Relative difference ((2020–2019)/2019 data) in the number of athletes running per week, mean and 95% CI of the relative difference in the running distance, and mean COVID-19 government response stringency index during 2020, weighted by the number of athletes in each country (see the labels on the vertical axes). At the bottom of the graph, the horizontal bars in the first line indicate a small effect size (between 0.2 and 0.5) and the horizontal bars in the second line of indicate a statistical difference (p < 0.001) between the corresponding weeks of 2019 and 2020 for the weekly running distance.

Average values across the year of the changes in the outcome variables for all countries and the ten most represented countries in the dataset are shown in Fig. 4 and graphs of the annual time series for number of athletes and running distance for Brazil are shown in Fig. 5.

Figure 4 Changes in running training and the 2020 stringency index for different countries.

Mean and standard error across the year of the decrease in weekly running distance and in the number of athletes running in 2020 compared to 2019 (in %) and of the 2020 stringency index for all countries (36,412 athletes) and for the ten most represented countries in the dataset. The horizontal lines in the graph represent the mean values of the data for all countries. Below each label on the horizontal axis, the percentage value of the number of athletes of each country in the dataset is shown.

Figure 5 Comparison between the weekly running training of 652 athletes from Brazil over the years 2019 and 2020.

(A) Number of athletes running per week by year. (B) Mean and 95% confidence interval (95%CI) of weekly running distance by year. (C) Relative difference ((2020–2019)/2019 data) in the number of athletes running per week, mean and 95%CI of the relative difference in the running distance, and mean COVID-19 government response stringency index during 2020 in Brazil (see the labels on the vertical axes). At the bottom of the graph, the horizontal bars in the first line indicate a small effect size (between 0.2 and 0.5) and the horizontal bars in the second line of indicate a statistical difference (p < 0.001) between the corresponding weeks of 2019 and 2020 for the weekly running distance.

The relative difference in the weekly running distance and number of athletes between the years 2019 and 2020 stratified by gender and age range for all athletes worldwide is shown in Fig. 6.

Figure 6 Comparison between the weekly running training of 36,412 athletes worldwide over the years 2019 and 2020 grouped by gender and age range.

Relative difference ((2020–2019)/2019 data) in the number of athletes running per week and mean and 95% confidence interval of the relative difference in the running distance (see the labels on the vertical axes). In each graph, the numbers in the upper left corner indicate the mean weekly running distance and in the upper right corner the number of athletes running per week in 2019 and 2020 grouped by gender and age range. At the bottom of each graph, the horizontal bars in the first line indicate a small effect size (between 0.2 and 0.5) and the horizontal bars in the second line of indicate a statistical difference (p < 0.001) between the corresponding weeks of 2019 and 2020 for the weekly running distance.

Discussion

We extracted information related to physical activity recorded in all days of 2019 and 2020 by web scraping a large social network for athletes on the internet. We hypothesized that compared to 2019, the training volume (measured by running distance and duration) for long-distance running in 2020 would be affected. Considering each year as a whole, the long-distance running training and the number of athletes running worldwide in 2020 in comparison with 2019 do not seem to have been much affected by the COVID-19 pandemic, since we observed 3.6% reduction in the total number of athletes running (with a 6.5% reduction on athletes running per week) and an average reduction of only 7% for running volume from the previous year. Due to the intra-subject design of our study, it would not be possible to observe an increase in the number of athletes in 2020 and, in fact, a decrease in this number was expected, for example, due to injury or any other factor that would lead athletes not to run in 2020. Although the COVID-19 pandemic could be one of these factors, it is not possible to identify the exact cause of this decrease solely from the total number of athletes who ran. However, there were large variations throughout 2020, reaching 35% lower running distance and 16% fewer athletes running per week in September 2020 compared with the same period in 2019 and at least the large drop in in the running data seem to be linked to the start of the preventive measures against COVID-19. These variations in the outcome variables suggest greater irregularity in running training, which itself can be a concern for potential injuries (DeJong, Fish & Hertel, 2021).

In 2020 and considering the data for all athletes, it can be observed that the start of the decrease in running training seems to coincide with the start of the preventive measures against COVID-19 in early March, as highlighted by the 2020 stringency index (Ritchie et al., 2020). But in the second quarter of 2020, there was a break in this relationship. Although the preventive measures remained, we observed not only a return to the running training levels, but even an increase relative to the same period in 2019. It is only after the second half of 2020 that the running training decreases again, probably due to the preventive measures remaining in force for a longer time. During the final months of 2020, running training returned 2019 levels, although the level of restrictions did not change. The reasons for this running rebound are not clear. We speculate that athletes relaxed their own strict adherence to public policies, or found ways to run while coping with the preventive measures; for example, running alone, wearing a mask, or running at times when there were fewer people on the streets. Since similar patterns of variation were observed for the average weekly running distance and duration, this suggests that there was no change in the runners’ pace in 2020 compared with 2019.

In the first half of 2020, compared with the same time period in 2019, running volume decreased in the first quarter, but then increased in the second quarter, perhaps as a compensation for the decrease in running training in the previous month. These results are in line with other articles and reports about general physical activity (Fitbit, 2020; Strava, Inc 2020; Tison et al., 2020; Stockwell et al., 2021). However, the company Strava, from which the data for this study were extracted, describes in their 2020 annual report (Strava, 2020) a 14.7% increase in the total duration of physical activity by their users, worldwide. In particular, the 2020 report by Strava does not describe a decrease in physical activity in the second half of the year as we observed for running. Possible reasons for this apparent contradiction are that we only analyzed data from long-distance running and from the same athletes who were already running in 2019, without including new users who started using the Strava app in the year 2020. Strava described in its 2020 report an increase of about 40% in the number of users in 2020, and it was data from these users that were used to describe overall physical activity in 2020. Note that we report the amount of physical activity per athlete; if we used the same metric for the Strava data, we would likely also find an average decrease in physical activity per athlete (an increase of only 14.7% in physical activity volume after a 40% increase of users). Another important aspect is that the cancellation of all six World Marathon Majors (Abbott World Marathon Majors , 2020) in 2020 (as well as other, similar public events) may have influenced the athletes in this dataset more than general users of Strava, since the athletes in this dataset may train with the incentive to run a marathon. Indeed, a survey of 1,147 runners from 15 countries showed that the COVID-19 pandemic was associated to a decrease in motivation and an increase in injury risk (DeJong, Fish & Hertel, 2021). Curiously, two studies have specifically investigated the effect of the COVID-19 pandemic on the practice of running and both studies found an increase in running volume during the pandemic (DeJong, Fish & Hertel, 2021; Chan et al., 2022). The study by the DeJong and collaborators (based on data of 1,147 runners across 15 countries) found an average increase of 1.4 km (from 39.6 to 41.0 km) in weekly running distance (DeJong, Fish & Hertel, 2021). The study by Chan and collaborators (based on data of 65 runners across 5 countries) found an average increase of 3 km (from 25 to 28 km) in weekly running distance after the start of COVID-19 restrictions (Chan et al., 2022). We observed an average decrease of 2.2 km (from 29.2 to 27.0 km). The reason for this discrepancy is unclear; however, an important methodological difference from the present study to the other two is that both studies recruited subjects after the onset of the COVID-19 pandemic (retrospectively), whereas the data from the present study were collected by web scraping, without recruitment. It is possible that runners who agreed to participate in the other two studies (after the investigation period ended) were less affected by the pandemic at that time than runners who did not volunteer.

Running training in different countries were unevenly affected by the COVID-19 pandemic and restriction measures. Among the ten most represented countries in the dataset, running training in Brazil seems to have been more affected than in these other countries and than the average of the 130 countries present in the dataset; there were greater decreases both in the number of athletes and in the distance covered.

The present article is the first to report an open dataset and an analysis of the effects of the COVID-19 pandemic on the practice of long-distance running, considering the entire 2020 year, including comparison of data from a large worldwide sample of individuals between the year 2020 and the year 2019, in the context of a within-subject design. By comparing the data from 2019 with those from 2020, we were able to better control for the effects of seasonality, and the within-subject design reduced errors related to inter-individual differences. Considering that the number of runs and running volume varied widely among athletes, future analyzes of this dataset could be stratified by running frequency and volume and the athlete’s performance level, e.g., using their marathon finishing time grouped by age as an indicator, as in Clermont et al. (2019).

An important aspect of the results presented is that there is great between-subject variability in the data, as indicated, for example, by their wide statistical distributions (see Figs. 1 and 2). The large between-subject variability contributed to the small effect sizes observed in the differences found in this study and may also explain part of the disparity in relation to the literature on this topic. Looking at all the world’s athletes in the dataset, overall running training behavior does not appear to be affected by gender and age group; since we observed patterns similar to those described for all athletes.

Our study contains limitations that may have affected the results. The dataset analyzed is a convenience sample, and may not be representative of all users of a specific application to record running activities. As such, running activities were self-reported and may contain errors due to the use of different devices to measure activity or the method of recording and loading the information on the platform. Other information, such as gender and age group, was also self-reported, and as with running training information, it is not possible to verify the veracity of this information. The use of wearable technology for a stride-to-stride analysis of running biomechanics in an outdoor environment has been shown to be feasible (Clermont et al., 2019), but the current data is limited to information on total distance and time covered. Technically, we cannot affirm that the running activities were in fact long-distance running (the athlete could be training for sprint running), although all athletes had run a marathon before and their weekly average running volumes suggest they were performing long-distance running training.

Conclusions

In conclusion, in terms of running differences in 2020 compared with 2019, we found wide variations in running volume (distance and duration) and the number of athletes running, throughout 2020, which are likely related to the COVID-19 pandemic. As for the total running volume, the observed decreases of up to 7.5% in the outcome variables related to running training in 2020 could also be attributed to the COVID-19 pandemic, but other factors such as injury, illness or lack of interest, may also have contributed to these decreases. Given the importance of being physically active for a better quality of life, the population and policy makers must find ways to ensure the practice of physical activity during a pandemic like the one we are experiencing. We view this study as exploratory and, considering that the dataset and the Jupyter notebooks for the analyzes described here are openly available, we invite the reader to reproduce and extend the investigation to better understand the behavior of running training.

Additional Information and Declarations

Competing Interests

Author Contributions

Data Availability

The authors declare there are no competing interests.

Leonardo A. Afonseca, Renato N. Watanabe and Marcos Duarte conceived and designed the experiments, performed the experiments, analyzed the data, prepared figures and/or tables, authored or reviewed drafts of the paper, and approved the final draft.

The following information was supplied regarding data availability:

The data is available at Figshare: Afonseca, Leonardo; Watanabe, Renato Naville; Duarte, Marcos (2021): A public dataset on long-distance running training in 2019 and 2020. figshare. Dataset. https://doi.org/10.6084/m9.figshare.16620238.v4.

The source code and Jupyter Notebooks are available at GitHub: https://github.com/BMClab/covid19/.

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
