# Peer review of "A worldwide comparison of long-distance running training in 2019 and 2020: associated effects of the COVID-19 pandemic"

_PeerJ, doi:10.7717/peerj.13192_

## Round 0.1 · original submission · Major Revisions

· Academic Editor

Major Revisions

Although considering interesting the article, reviewers required some changes.

·

Basic reporting

I think that this section is enough.

Experimental design

Detailed information is given in the method section of the article.

Validity of the findings

The results of the study contribute to the literature

·

Basic reporting

This study employs web scraping, a relatively new tool in the arsenal of sports scientists, to investigate various aspects of training behavior. The authors have generated search criteria for the largest SoMe database of training and exercise based on a practical benchmark of having run one 6 defined “major” international marathons (3 USA, 2 Europe, 1 Japan). The running behavior (frequency, running distance, running duration) of this cohort was compared across 2019 and 2020 to quantify changes in exercise behavior chronologically aligned with the C-19 pandemic and social distancing restrictions imposed by most of the world.
The main results presented for this global natural experiment were very modest reductions in running frequency and average running duration or distance, but large variation within the comparison years. Both decreased running activity (early lockdown) and increased running activity relative to 2019 were observed, despite similar restrictions. At a broad level, the data suggest that runners either stopped observing the social distancing rules or found ways to train more within the restrictions.
This study represents a reasonably concise “first pass” analysis of a large cohort of runners. The authors have made necessary delimitations both in terms of the data search criteria and the specific analyses performed and presented. The available data from this study would be ripe for additional analyses by scientific peers, thereby further enhancing the value of the methodological approach. For example, my own interests as an investigator of endurance training practice would be to further restrict the cohort to runners who had trained at higher frequencies/volumes in 2019 and then repeated the comparison of training characteristics among “highly motivated” runners who were left without major objective “target races” in 2020.
I found the study to be understandably presented. The authors have contextualized this study within the background of the pandemic and the relatively large set of studies that have been published in a short time around the issue of physical activity changes during the Covid-19 Pandemic. This is the first study to the authors’ knowledge that compared data for all of 2020 with pre-pandemic data. The literature list is short but includes a review article of 66 published studies investigating physical activity during the pandemic. Only 4 of these have used objective data (GPS, accelerometry), making this study a unique contribution when combining objective data representing the entire first year of the pandemic (2020).

My general assessment of the manuscript is positive, and I understand the challenges and opportunities presented by this methodological approach. In some ways, this is a “proof of concept” paper demonstrating the potential utility of web-scraping approaches in the physical activity and sports field (and other social data)
Like all social behaviours, running behaviour is not normally distributed, even within “runners”. Running volume exhibits a long-tail distribution and it would have been interesting if the authors could show this distribution for 2019 and for 2020 (or by quarters). It seems possible that runners on different positions within this distribution would show different behaviors when faced with both social distancing and cancellation of running events. In my opinion, this study would be much more interesting if such data were included here.

Experimental design

No additional comment.

Validity of the findings

The authors make all data available via the Github repository and the validity of the findings can be independently evaluated by other investigators.

It would also be useful to know some background regarding the growth of Strava over the timeframe. Can the authors provide some indicators for how representative was 2019 data alone of “Pre-Covid” conditions?

The manuscript would be improved by reporting the range of years across which the cohort had run at least one major and reported it on Strava?

The average number of training sessions reported on Strava per runner works out to 294 activities across the 2 years. Please provide the frequency distribution for this metric.

I find the conclusions to be fully consistent with the data analysis presented.

Additional comments

Minor editing suggestions:


L39 ….activity guidelines associated with a reduced risk…..
L41 …suggesting that overall physical activity decreased…..
L42 ….when preventive measures were adapted against COVID-19, such as social distancing and isolation (delete began to be applied, compared to previous periods)
L48 …..review, quantified physical activity in the first half of…..
L68 Our hypothesis was………would be reduced.
L74 Data extraction was performed……
L97 …to search all associated web pages. (delete the rest of this sentence).
L98 and who did not provide information on gender and age group.
L99….we have successfully identified a cohort of distance runners…..
L118 The original dataset analyzed in this study included 37,595 athletes and 14,644,391 separate activity bouts performed in 2019 and 2020.
L156 …differences were not normally distributed……
L166 …the 95%CI ranges were calculated….
L174 …we conducted non-parametric….
L271 During the final months of 2020, running training returned to 2019 levels…..
L276 The reasons for this running rebound are not clear. We speculate that athletes relaxed their own strict adherence to public policies, or found ways to ……

Reviewer 3 ·

Basic reporting

See notes below

Experimental design

See notes below

Validity of the findings

See notes below

Additional comments

This study analyzed publicly available training records for over 36k athletes, comparing their weekly running volume and duration between 2019 and 2020 to assess the effect of COVID-19. This study used a within-subject comparison and repeated the comparison for each week to counter seasonal and individual variations. Differences in weekly training volume and duration were found in March (onset of restrictions amid the pandemic) and the largest differences were recorded in September.

Overall
Strength:
This study made use of an open dataset with real-world training record, which is appropriate for this type of studies (i.e., understanding changes over a period of time). The authors reported details on how participants were selected, using available data to set and screen for participants based on the inclusion/exclusion criteria. They have also performed
sub-group analysis (age-group and gender) and provided details for how they obtain (and estimate) the age group of athletes based on available records. The methods used in this study could be referenced in future studies.

Limitations:
While I understand the difficulty in screening participants based on available data, the inclusion criteria have limited the generalizability of the findings, only runners who have completed a marathon (out of the 6 major marathons) were included.

The authors reported a 3.6% reduction in athletes training (and up to 16% in certain weeks). This decrease is inherent, as they have excluded runners who did not run in 2019 but in 2020. There can either be no change or a decrease, and without a figure for comparison (e.g., 2018 vs. 2019), it is impossible to interpret if the change is likely due to the pandemic or not, as there are other possible reasons for runners to stop running (i.e. injury).

There is a lack of discussion on the significance of the study. Results of the analyses were reported without connection to COVID-19. The Stringency Index could be used to help better understand the effect of COVID-19. If possible, the authors could analysis the difference between locations with high Stringency Index to see if there is any relationship between the severity and the changes observed in running volume.

Introduction
I suggest the authors elaborate on the shortcomings/limitations of previous studies to further strengthen the need of the current study. It was mentioned within the introduction that previous studies relied on questionnaires (line 47) and Strava included new users in 2020 (line 55 -57), but they did not explain why that would be substandard for answering the research question. The knowledge gap has to be better defined and how the current study design is more appropriate should also be explained.

The reason why long-distance running was used as a measure of physical activity was only briefly mentioned in the last paragraph of the introduction (line 65 – 76). This study is primarily study changes among long-distance runners, the authors should consider shifting the focus onto long-distance running rather than physical activity. There are published studies on how running habits changed during the pandemic, the authors should mention that instead of only focusing on physical activity within the first 2 paragraphs of the introduction.

You may wish to include this recent publication from Chan et al: (https://www.frontiersin.org/articles/10.3389/fspor.2021.812214/abstract) in the Intro and Discussion.

Methods
The methods section included enough details and was written clearly enough for reader to follow. The study design and analyses can potentially be referenced in future studies.

Results
The authors may want to calculate Age-Grade levels for these runners, based on their finish time for the marathons and their age (Clermont et al., J Appl Biomech. 2019:1-9. doi: 10.1123/jab.2018-0453). This may help put the results in perspective for the reader and provide further insight into the results. The age-grade for the runner may also provide new sub-groups for comparison.

Section 3.1 might fit better in the methods or as supplementary material.
If possible, present number (value and % changes) in a tubular format for clarity.

Figure 1 and 2 both contains a lot of information and look very busy to a reader. The authors may consider:
1. Putting the stringency index in a subplot above the training volume/duration.
2. Removing the number of athletes (red line) from either one of the figures. They are the same and do not add more value when presented on both figures.

The blue shaded area (assuming it is the 95% CI of the change in weekly running volume) is only visible on graph D and G.
There are 2 rows of light blue bars on each graph, are they both indicating statistical differences?

Line 236 - 237: speculation and interpretation of the results would fit better in the discussion.
Line 237: Statistical test used for analyzing association between the stringency index and running volume was not mentioned. If the association is determined by observation, please specify.

Discussion
Overall, the discussion can use more elaboration on the practical implication of the study. Results are presented and repeated in the discussion, but there is a lack of comparison with other studies or other physical activities. The significance of the study to future research and athletes should also be discussed.

Line 260 – 264: “running training and number of athletes running … do not seem to have been much affected by COVID-19…” this interpretation needs to be supported.

The inclusion criteria (i.e., only marathon runners) should be justified. I understand the need for the delimitation (as mentioned in line 95). I suggest discussing the feasibility (and difficulty) of including recreational (non-competing) runners within the discussion.

The authors have averaged (weighted) the stringency index based and presented the overall change of training volume between 2019 and 2020. A location-specific (or grouping based on stringency index) comparison might further reveal the effects of COVID-19.

Line 293 – 297: the change in motivation and the effect on race cancellations has been studied qualitatively. The authors may reference other studies (i.e., https://doi.org/10.1371/journal.pone.0246300) to support their speculation.

Conclusion
The association between irregularity in running training and injury should be mentioned in introduction and discussion.

---

## Round 0.2 · accepted · Accept

· Academic Editor

Accept

The revisions are fine, and the article can be accepted.

Reviewer 3 ·

Basic reporting

The authors have done a nice job addressing our concerns and suggestions. One last item that needs to be addressed is on line 346 of the Discussion which states: "...without recruitment (almost in a prospective way). " Please eliminate "(almost in a prospective way)." as this is not appropriate nor correct.

Experimental design

N/A

Validity of the findings

N/A

Additional comments

N/A